# Interactions between microbiota and uterine corpus endometrial cancer: A bioinformatic investigation of potential immunotherapy

Samia S. Alkhalil[1]*, Taghreed N. Almanaa[2], Raghad A. Altamimi[2], Mohnad Abdalla[3], Amr Ahmed El-Arabey [4,5,6]*

1 Department of Medical Laboratory Sciences, College of Applied Medical Sciences, Shaqra University, Alquwayiyah, Riyadh, Saudi Arabia, 2 Department of Botany and Microbiology, College of Science, King Saud University, Riyadh, Saudi Arabia, 3 Pediatric Research Institute, Children's Hospital Affiliated to Shandong University, Jinan, China, 4 Department of Pharmacology and Toxicology, Faculty of Pharmacy, Al-Azhar University, Cairo, Egypt, 5 Center of Bee Research and its Products (CBRP), Unit of Bee Research and Honey Production, King Khalid University, Abha, Saudi Arabia, 6 Applied College, King Khalid University, Abha, Saudi Arabia

* salkhalil@su.edu.sa (SSA); amrel_arabey@azhar.edu.eg (AAE)

**Funding:** The author(s) received no specific funding for this work.

## Abstract

Microorganisms in the gut and other niches may contribute to carcinogenesis while also altering cancer immune surveillance and therapeutic response. However, determining the impact of genetic variations and interplay with intestinal microbes' environment is difficult and unanswered. Here, we examined the frequency of thirteen mutant genes that caused aberrant gut in thirty different types of cancer using The Cancer Genomic Atlas (TCGA) database. Substantially, our findings show that all these mutated genes are quite frequent in uterine corpus endometrial cancer (UCEC). Further, these mutant genes are implicated in the infiltration of different subset of immune cells within the Tumor Microenvironment (TME) of UCEC patients. The top-ranking mutant genes that promote immune cell invasion into the TME of UCEC patients were PGLYRP2, OLFM4, and TLR5. In this regard, we used the same deconvolution of the TCGA database to analyze the microbiome that have a strong association with immune cells invasion with TME of UCEC patients. Several bacteria and viruses have been linked to the invasion of immune cells, such as B cell memory and T cell regulatory (Tregs), into the TME of UCEC patients. As a result, our findings pave the way for future research into generating novel immunizations against bacteria or viruses as immuno-therapy for UCEC patients.

## 1- Introduction

Gut flora is a biological system created in the digestive tracts by microbial cells of typical pro-karyotic and eukaryotic origin. The number of bacteria in an adult male's gastrointestinal track is approximately 100 trillion, which is roughly double the size of our cells [1]. The human digestive system is home to a diversified microbial population that interacts intimately with the mucosal immune system. Gut microbiota is important in sustaining host health

**Competing interests:** The authors declare that they have no known competing financial interests or personal relationships that could have appeared to influence the work reported in this paper.

because it may give nutrients, manage energy balance, influence immunological response, and provide protection against infections [2]. Intestinal bacteria are essential for immunological and metabolic balance as well as pathogen defense [1]. Individuals have their unique interplay of microbiome that is governed by their DNA, and microbes are initially introduced to a person as a newborn [3]. Intestinal microbe genetic alterations can be condition-specific and might potentially provide a medical forecast for the preliminary phases of disorders [3]. Dysbiosis has been associated to a number of inflammatory diseases and infections [1]. Gut microbial dysbiosis has been linked to a variety of disorders ranging from irritable bowel syndrome to cancer [4]. A recent study found that the host TP53 regulates symbiosis and immunological homeostasis via sialic acid metabolism. Disrupted sialic acid metabolism caused by a TP53 mutation might be used by particular gut microbiome components, resulting in dysbiosis and inflammation. Manipulation of sialo metabolism may thus provide an effective treatment approach for dysbiosis induced by TP53 mutation, inflammation, and, eventually, cancer [5]. Tumor cell spread is mediated by contact and cross-talk between cancer cells (seed) and the host organ (soil), and the premetastatic niche is made of aberrant proteins from the extracellular matrix and immune cells accumulating in organs targeted [6]. Tumor-infiltrating immune cells are critical in the fight against cancer. Malignant cells, can elude the immunological reactions and promote tumor development, metastasis, and resistance to treatment through invasion of characteristic subtypes of immune cells [7]. Emerging evidence suggests that bioinformatic investigations of The Cancer Genomic Atlas (TCGA) human data are an intriguing way to grasp the complicated interactions within TME. Several studies have found a link between mutant genes and immune cells invasion inside TME, such as TP53 status orchestrating macrophage infiltration in stomach cancer (STAD) and ovarian cancer (OV) patients [8–10]. Further, TP53 status was significantly linked to CD8+ T cells invasion in the TME of uterine corpus endometrial carcinoma, head and neck squamous cell carcinoma and lung adenocarcinoma patients [10]. Several reports demonstrate that colorectal cancer (CRC) start and progression are influenced by gut microbial dysbiosis via networks with the innate immune system of the host. The microbiota is a promising target in regulating immunotherapy responses in preclinical CRC models because of the tight relationship that exists between the gut microbiota and anticancer immune responses [11]. However, there is no evidence of a relationship between gene mutations that cause aberrant gut flora and many types of cancer. Hence, in the present proposal by utilizing bioinformatic analyses of the TCGA database, we designed to assess the prevalent mutations genes that generated abnormal gut flora and explore their frequency in different types of malignancies. Then, depending on the frequency of these mutations, we selected the most relevant malignancy to evaluate their influence on the invasion of various immune cells within TME. In addition, the impact of recognized bacteria and viruses on immune cells infiltration within TME was being analyzed.

## 2- Material and methods

The mammalian phenotype browser was used in this investigation to identify gene mutations causing the aberrant gut flora balance phenotype in transgenic mice from the mammalian phenotype ontology via the gene-phenotype associations dataset [12–14]. Following that, we employed tumor immune estimation resources 2 (TIMER2) as a complete resource for analyzing immune infiltrates in cancers of all types. TIMER2 was used to perform bioinformatics analysis using multiple modules, including the module for gene mutation, which evaluates gene expression variations across several mutation statuses of the recognized mutant genes that generated abnormal gut flora balance. The log2 fold changes in each gene's differential expression were used in the graph of each malignancy [15–17]. The mutation module was

then used to examine and visualize the influence of gene alterations on immune cells infiltration including T cell CD4$^+$ memory resting, T cell CD4$^+$ memory activated, M1 Macrophage, T cells regulatory (Tregs), Monocytes, Eosinophil, T cell follicular helper T cell gamma delta, T cell CD8$^+$, M0 Macrophage, NK cell activated, Myeloid dendritic activated, resting myeloid dendritic, M2 Macrophage, B cell memory and activated mast cell activated across various cancer kinds and immune cell types at the same time. This module took advantage of TIMER2 to demonstrate the TCGA algorithm using CIBERSORT. The bar plots display the frequency of gene mutations for each TCGA cancer category. TIMER2.0 illustrates a violin plot displaying the spread of immune infiltration in mutant versus wild-type tumors [15–17]. Human genetics has a substantial influence on the composition of cancer microbiome. Cancer-mbQTL was constructed using the NodeJS 8.10.0 framework, on a variety of web browsers, including Google Chrome, Firefox, Internet Explorer, and macOS Safari [18]. We used the Cancer-mbQTL database to discover immune cells from CIBERSORT deconvolution of TCGA data that correlate with the abundance levels of microbes in tumor tissues of UCEC patients (designated as cancer microbiome quantitative trait loci, cancer-mbQTL) [18]. Given the importance of microbiota-immune system interactions in tumor growth by inhibiting local antitumor immunity, we calculated the connections between microbiome and immune cell infiltration, resulting in prospective microbiota-based biomarkers for cancer immunotherapy. In this regard, we used Spearman correlation to determine the relationship between microbial abundance and immune cell infiltration in uterine corpus endometrial cancer using CIBERSORT method. Associations with FDR < 0.05 were deemed statistically significant [18]. Kraken-derived normalized microbial abundances of TCGA patients were collected from the Cancer Microbiome. This comprehensive mbQTL database aids in successfully evaluating the influence of gene mutations on cancer microbiome characteristics, establishing a new paradigm for understanding the function of risk genetic mutations in human malignancies [18].

## 3- Results

### 3.1- Thirteen gene mutations cause the abnormal gut flora balance phenotype and are common in uterine corpus endometrial cancer

Using the mammalian phenotype browser, we found thirteen gene mutations (IL18, NLRP6, OLFM4, TLR5, PGLYRP1, AICDA, IKZF1, PGLYRP2, PGLYRP3, PGLYRP4 MYD88, PYCARD and REG3G) from the mammalian phenotype ontology gene-phenotype relationships dataset that cause an abnormal gut flora balance phenotype in transgenic mice (**Table 1**). TIMER2 was used to evaluate the frequency of these thirteen gene mutations among 30 sorts of cancer from the TCGA database. Substantially, our investigation revealed that five of them (IL18, NLRP6, OLFM4, TLR5, and PGLYRP1) had the highest-rated mutation in UCEC (**Fig 1A–1E**). Among these top-rated gene alterations in UCEC, TLR5 mutation (**Fig 1D**) had the highest proportion about 7.7% and IL18 mutation (**Fig 1A**) had the lowest percentage around 1.5%. Our findings also revealed that AICDA, IKZF1, PGLYRP2, PGLYRP3, and PGLYRP4 were the second most common gene mutations in UCEC (**Fig 2A–2E**). The PGLYRP2 mutations had the largest proportion of mutations, accounting for approximately 4% of these second-ranked alterations (**Fig 1C**), whereas PGLYRP3 mutations had the lowest rate, accounting for around 2.6% (**Fig 1D**). In contrast, MYD88, was the third most prevalent mutation in UCEC across 30 types of cancer (**Fig 3A**), with a frequency of roughly 1.7%. PYCARD was the fourth most prevalent mutation in UCEC (**Fig 3B**), with a 0.6% incidence rate. REG3G mutation was the fifth most prevalent alteration in UCEC, accounting for approximately 2.2% (**Fig 3C**).

**Table 1. The mammalian phenotype ontology for gene phenotype associations dataset comprises 13 gene mutations that generate the abnormal gut flora balance phenotype in transgenic mice.**

| Symbol | Name |
|--------|------|
| AICDA | activation-induced cytidine deaminase |
| IKZF1 | IKAROS family zinc finger 1 (Ikaros) |
| IL18 | interleukin 18 |
| MYD88 | myeloid differentiation primary response 88 |
| NLRP6 | NLR family, pyrin domain containing 6 |
| OLFM4 | olfactomedin 4 |
| PGLYRP1 | peptidoglycan recognition protein 1 |
| PGLYRP2 | peptidoglycan recognition protein 2 |
| PGLYRP3 | peptidoglycan recognition protein 3 |
| PGLYRP4 | peptidoglycan recognition protein 4 |
| PYCARD | PYD and CARD domain containing |
| REG3G | regenerating islet-derived 3 gamma |
| TLR5 | toll-like receptor 5 |

## 3.2- Status of IL18, NLRP6, OLFM4, and TLR5 considerably orchestrate immune cells invasion within the tumor microenvironment of UCEC

Our bioinformatics analysis of the TCGA database used the TIMER2 mutation module to investigate the effect of IL18, NLRP6, OLFM4, and TLR5 status on immune cells infiltration within the TME of UCEC. Notably, our data showed that IL18 status has a significant impact on the infiltration of T cell CD4$^+$ memory activated (**Fig 4A**), T cell CD4$^+$ memory resting (**Fig 4B**), and M1 macrophages (**Fig 4C**). Furthermore, the status of NLRP6 influences the invasion of T cells regulatory (Tregs) (**Fig 4D**), monocytes (**Fig 4E**), eosinophils (**Fig 4F**), T cell follicular helper (**Fig 4G**), and T cells gamma delta (**Fig 4H**). The status of OLFM4 indicated a significant contribution to the invasion of TME immune cells of UCEC patients, including T cell CD8$^+$, T cell CD4$^+$ memory resting, T cells regulatory (Tregs), monocyte, M0 macrophage, M1 macrophage, NK cell activated, and T cell follicular helper (**Fig 5A–5H**). Similarly, TLR5 status plays an important role in the TME of UCEC patients by controlling the invasion of T cell CD8$^+$, T cell CD4$^+$ memory activated/ resting, M1 macrophage, T cell follicular helper, T cell gamma delta, and myeloid dendritic cell activation (**Fig 6A–6G**).

## 3.3- The status of PGLYRP1, AICDA, IKZF1, PGLYRP2, PGLYRP3, PGLYRP4, REG3G, MYD88 and PYCARD in UCEC patients regulate immune cells infiltration

The TIMER2 mutation module was used in our bioinformatics analysis of the TCGA database to assess the effect of PGLYRP1, AICDA, IKZF1, PGLYRP2, PGLYRP3, PGLYRP4 REG3G, MYD88 and PYCARD status on immune cells infiltration into the TME of UCEC. The PGLYRP1 status revealed a particular infiltration of T cell gamma delta to the TME of UCEC patients (**S1 Fig**). AICDA status only regulates the invasion of activated M1 macrophages and myeloid dendritic cell activated (**S1B and S1C Fig**). In contrast, the status of IKZF1 regulates the invasion of M1 macrophages, M2 macrophages, T cell follicular helper, myeloid dendritic cell resting, and T cell gamma delta (**S1D–S1I Fig**). Considerably, the status of PGLYRP2 revealed the highest mutant gene, which orchestrates the invasion ten immunological cells in the TME of UCEC patients, including g T cell CD8$^+$, T cell CD4$^+$ memory activated, T cell CD4$^+$ memory resting, T cells regulatory (Tregs), B cell memory, M1 macrophages, myeloid

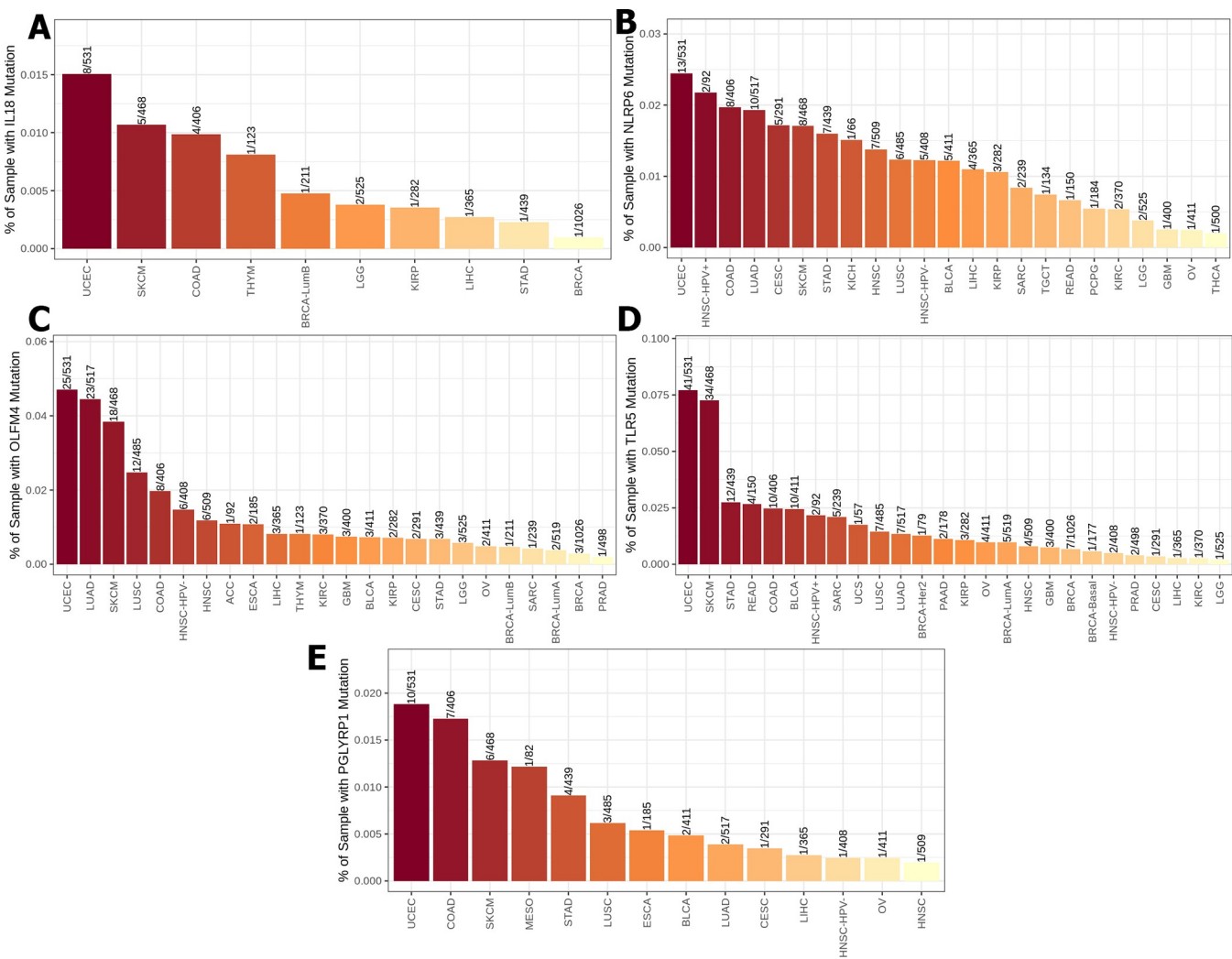

**Fig 1.** The percentage of TCGA database samples having IL18 (A), NLRP6 (B), OLFM4 (C), TLR5 (D), and PGLYRP1 (E) mutations across thirty types of cancer. UCEC is the most common malignancy linked to these mutations.

dendritic cell activated, mast cell activated, T cell follicular helper, and T cell gamma delta (**S2A–S2J Fig**). The PGLYRP3 status has a particular T cell follicular helper invasion into the TME of UCEC patients (**S3A Fig**). The status of PGLYRP4 controls the invasion of seven immune cells like TLR5 such as T cell CD8[+], T cell CD4[+] memory activated, T cells regulatory (Tregs), M1 macrophages, activation of myeloid dendritic cell, T cell follicular helper, and T cell gamma delta (**S3B–S3I Fig**). Following that, the status of REG3G was connected to the invasion of six immune cells, including T cell CD8[+], T cell CD4[+] memory activated/ resting, M1 macrophages and myeloid dendritic cell activated/resting (**S4A–S4F Fig**). MYD88 was also linked to the invasion of six immune cells in UCEC patients' TME, including T cells regulatory (Tregs), monocytes, M1 macrophages, myeloid dendritic cell activated, T cell follicular helper and T cell gamma delta (**S5A–S5F Fig**). Finally, the status of PYCARD regulates the infiltration of T cell CD4[+] memory activated, M1 macrophage, and T cell gamma delta (**S5G–S5I Fig**).

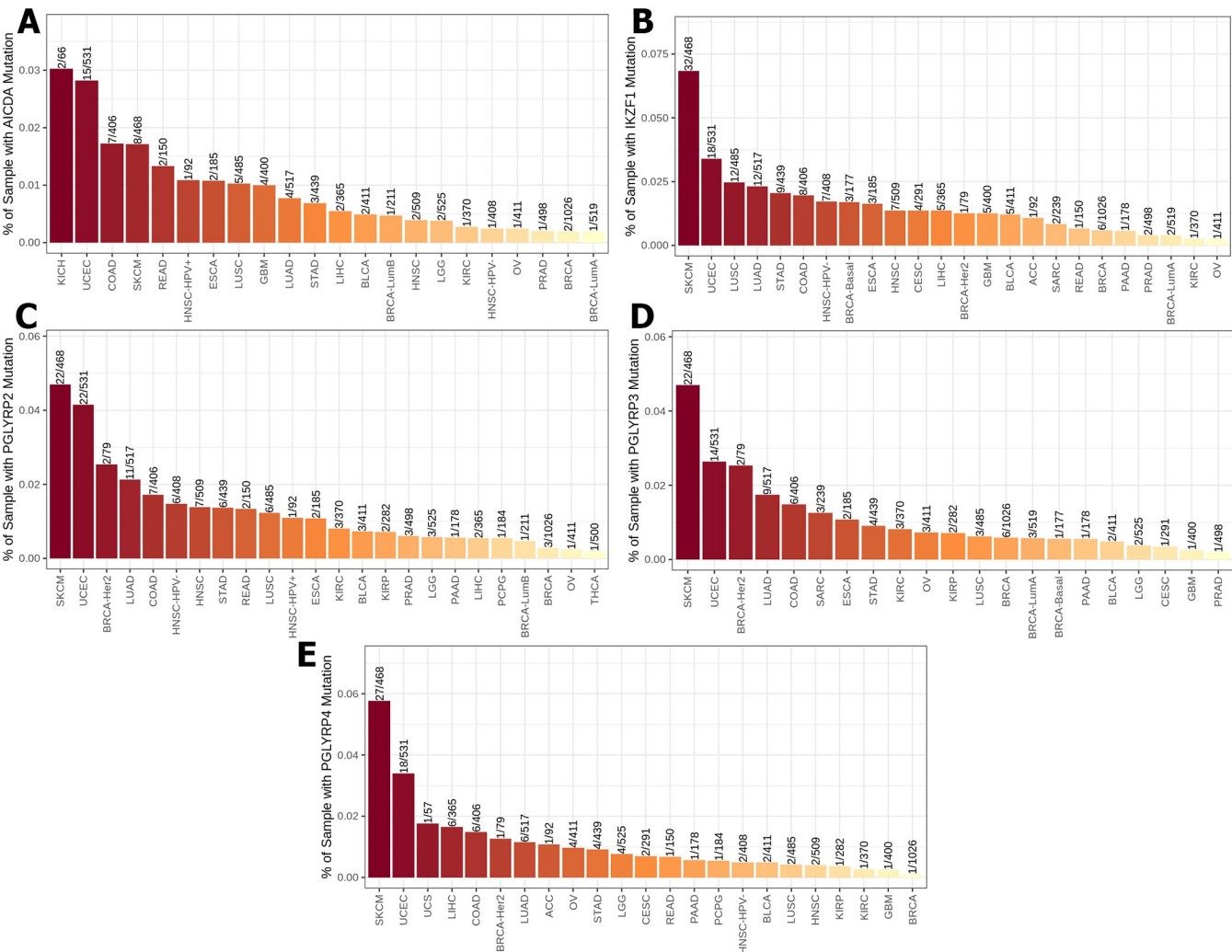

**Fig 2.** Across thirty types of cancer, the percentage of TCGA database samples with AICDA (A), IKZF1 (B), PGLYRP2 (C), PGLYRP3 (D), and PGLYRP4 (E) mutations. The second most prevalent cancer associated with these mutations is UCEC.

### 3.4- Cancer microbes show a considerably correlation with infiltration of T cells regulatory (Tregs), B cell memory, and M1 macrophage into UCEC patients' TME

Using the CIBERSORT deconvolution of TCGA database through the cancer-mbQTL data, we identified the microbes, including bacteria and viruses, discovered in UCEC. Following that, we investigated the impact of these microorganisms on immune cells infiltration into UCEC patients' TME. Next, we developed a Venn diagram tool to examine the intersection result between immune cells regulated by the thirteen mutant genes identified in our present proposal and which immune cells had a positive or negative Pearson correlation coefficient with the microbiome in UCEC patients. The intersection result demonstrates that both arms have B cell memory, T cell regulatory (Tregs), and M1 macrophages (**S1 File**). By using the same deconvolution CIBERSORT we selected the microbiome in UCEC patients that have positive or negative association with these immunological cells only (B cell memory, Tregs, and M1 macrophages). We identified the microbiome in UCEC patients that have a positive

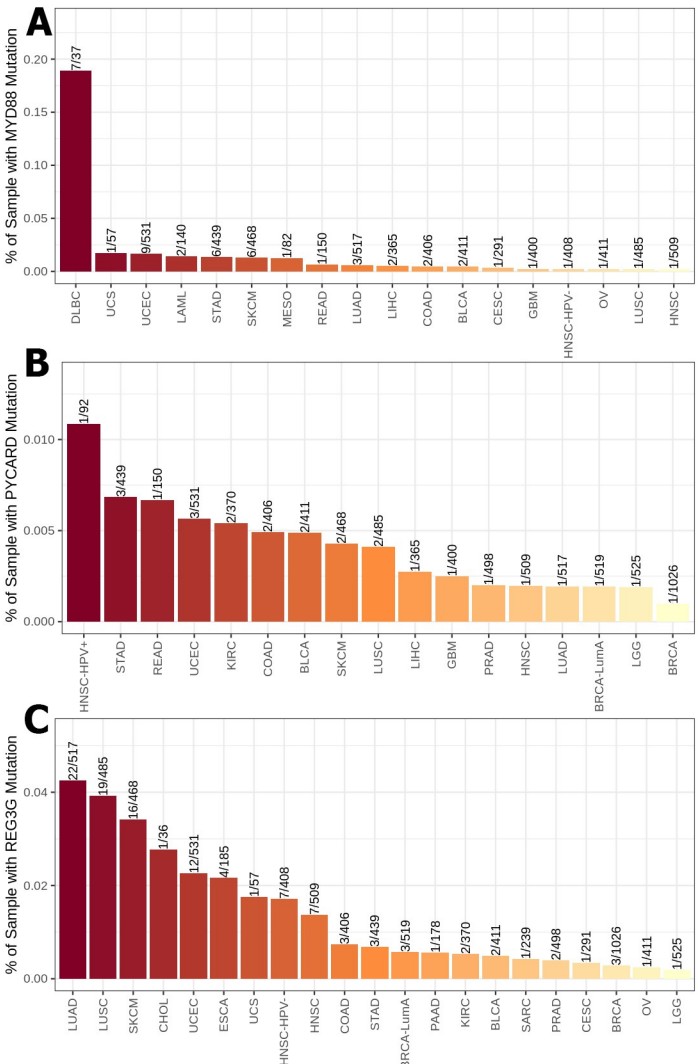

**Fig 3.** The proportion of TCGA database samples having MYD88 (A), PYCARD (B), and REG3G (C) mutations across thirty types of cancer. UCEC is the third most common malignancy linked with the MYD88 mutation, fourth with the PYCARD mutation, and fifth with the REG3G mutation.

or negative connection with these immune cells solely (B cell memory, Tregs, and M1 macrophages) using the same deconvolution CIBERSORT. Our findings showed that bacteria found in UCEC such as Proteobacteria (Bilophila), Bacteroidetes (Prevotella), Bacteroidetes (Riemerella), and Firmicutes (Parvimonas) have a negative Pearson correlation coefficient (-0.31, -0.3, -0.29 and -0.32 respectively) with Tregs invasion in TME (**Fig 7A–7D**). Furthermore, the data revealed that Bacteroidetes (Prevotella), Proteobacteria (Campylobacter), and Thaumarchaeota (nitrosopelagcus) have also a negative pearson correlation coefficient (-0.33, -0.29, and -0.32 respectively) with B cell memory invasion into UCEC patients' TME (**Fig 8A–8C**). In contrast, Firmicutes (Lachnoclostridium) and Bacteroidetes (Flammeovirga) exhibit a positive Pearson correlation coefficient (0.34 and 0.31) with M1 macrophage invasion (**Fig 9A and 9B**). Furthermore, both Proteobacteria (Ochrobactrum) and Proteobacteria (Citrobacter) exhibit a positive Pearson correlation coefficient (0.29 and 0.3) with Tregs migration into UCEC TME (**Fig 9C and 9D**). On the other hand, Polydnaviridae virus (Ichnovirus) displayed

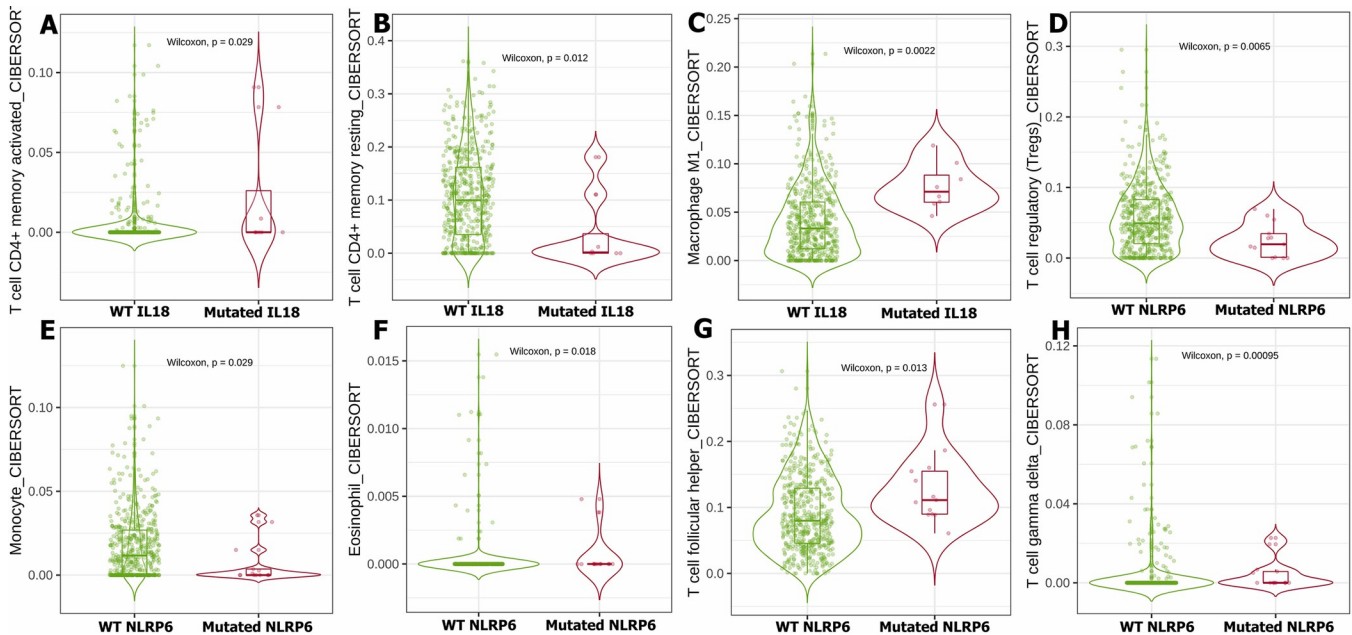

**Fig 4. The effect of IL18 and NLRP6 status on immune cell invasion with TME in UCEC patients was studied using the TCGA database.** In UCEC patients, the status of IL18 orchestrates the invasion of T cell CD4[+] memory activated (A), CD4[+] memory resting (B), and M1 macrophages (C). In UCEC patients, the NLRP6 status influences the infiltration of T cell regulatory (Tregs) (D), monocyte (E), Eosinophil (F), T cell follicular helper (G), and T cell gamma delta (H).

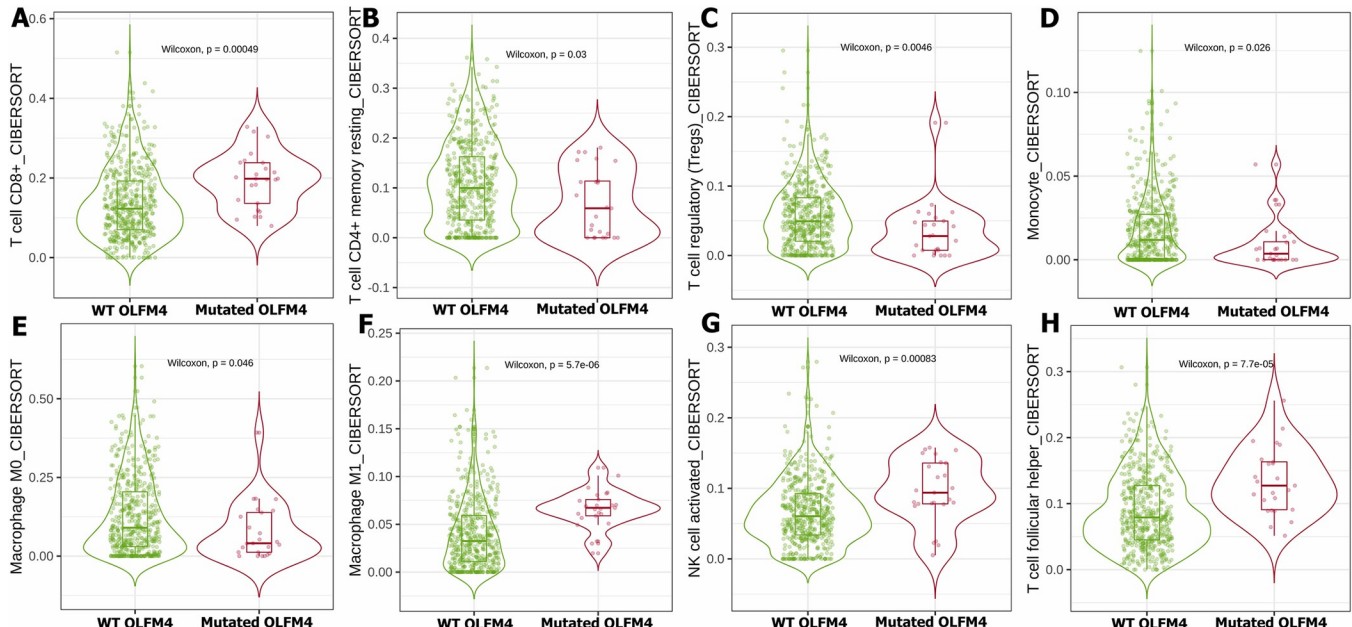

**Fig 5. The TCGA database was used to investigate the influence of OLFM4 status on immune cell invasion with TME in UCEC patients.** The status of OLFM4 orchestrates the invasion of T cell CD8[+] (A), CD4[+] memory resting (B), Tregs (C), monocytes (D), M0 macrophages (E), M1 macrophages (F), NK cell activated (G), and T cell follicular helper (H) in UCEC patients.

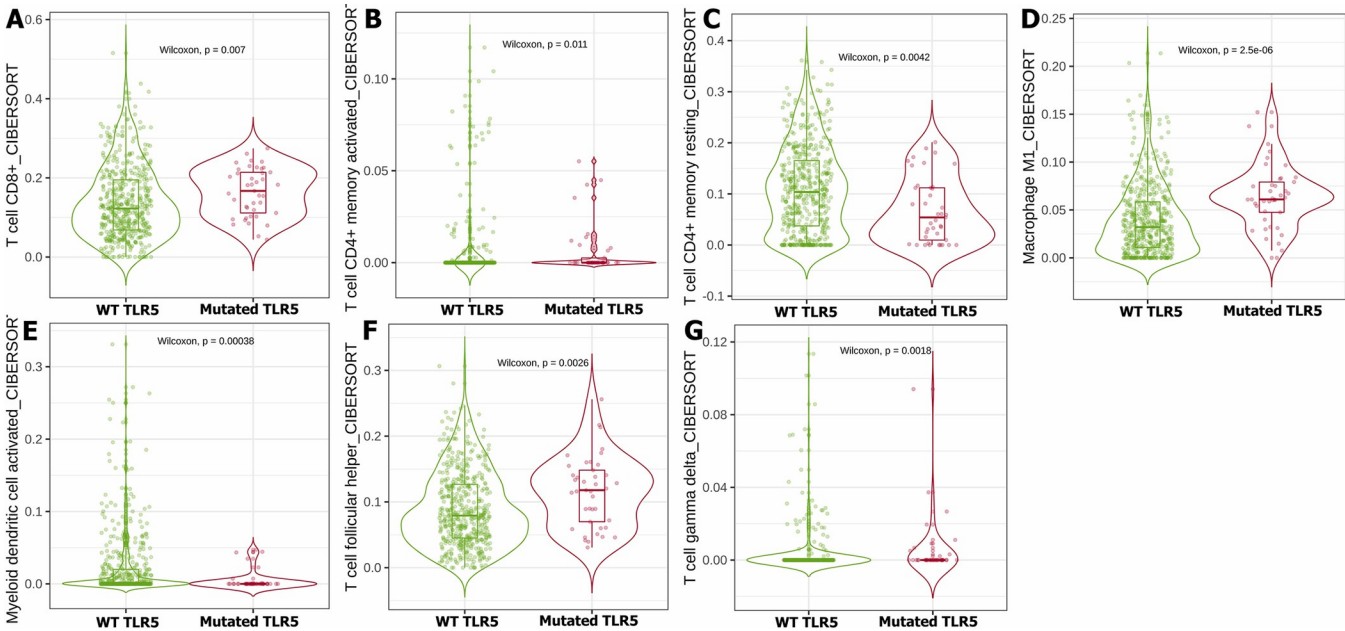

**Fig 6. The TCGA database was utilized to look at how TLR5 status affects immune cell invasion with TME in UCEC patients.** T cell CD8[+] (A), CD4[+] memory activated (B), CD4[+] memory resting (C), M1 macrophages (D), Myeloid dendritic cell activated (E), T cell follicular helper (F), and T cell gamma delta (G) invasion in UCEC patients is orchestrated by TLR5 status.

a negative pearson correlation coefficient (-0.34 and -0.32) with Tregs and B cell memory invasion into TME of UCEC (**Fig 10A and 10B**). Nonetheless, the Phycodnaviridae virus (Prymnesiovirus) displayed a Pearson correlation value (-0.32) with Tregs infiltration in UCEC patients (**Fig 10C**).

## 4- Discussion

The microbiome consists of bacteria, fungi, viruses, and their genes that naturally reside in and within the human body [19]. Cancer microbiome refers to microorganisms found in tumors, the species and number of which may have an impact on cancer formation, progression, and treatment response. Emerging evidence has shown the intricate role of microbiome in cancer genesis and treatment response [20]. In this sense, it was discovered that the altered probiotic E. coli Nissle strain produced nanobodies that would target immunological checkpoints including cytotoxic t lymphocyte-associated protein-4 and programmed death-ligand 1 intratumorally, as well as a possible tumor-reducing capacity [21]. Several studies have demonstrated that the crosstalk between the immunological system and the microbiota might be the foundation for the microbiome's function in cancer treatment. In this regard, more research into the link between host immunity and the microbiota is an appealing approach to understanding the etiology of treatment-related side effects [22]. Hence, we suggested in the current work to investigate the probable relationship between mutant genes generated aberrant gut flora and the frequencies of these mutant genes in different forms of cancer, as well as the clinical impact of these mutant genes on immune cells invasion within cancers. Besides, recognizing the potential microbiome that influence immune cells infiltration in malignancies. The bioinformatic analysis revealed thirteen mutant genes that generated aberrant gut flora. Researchers established the link between inflammatory bowel disease (IBD) in humans and colitis in animals to homeostasis disturbed by studying the instabilities in microbiome and

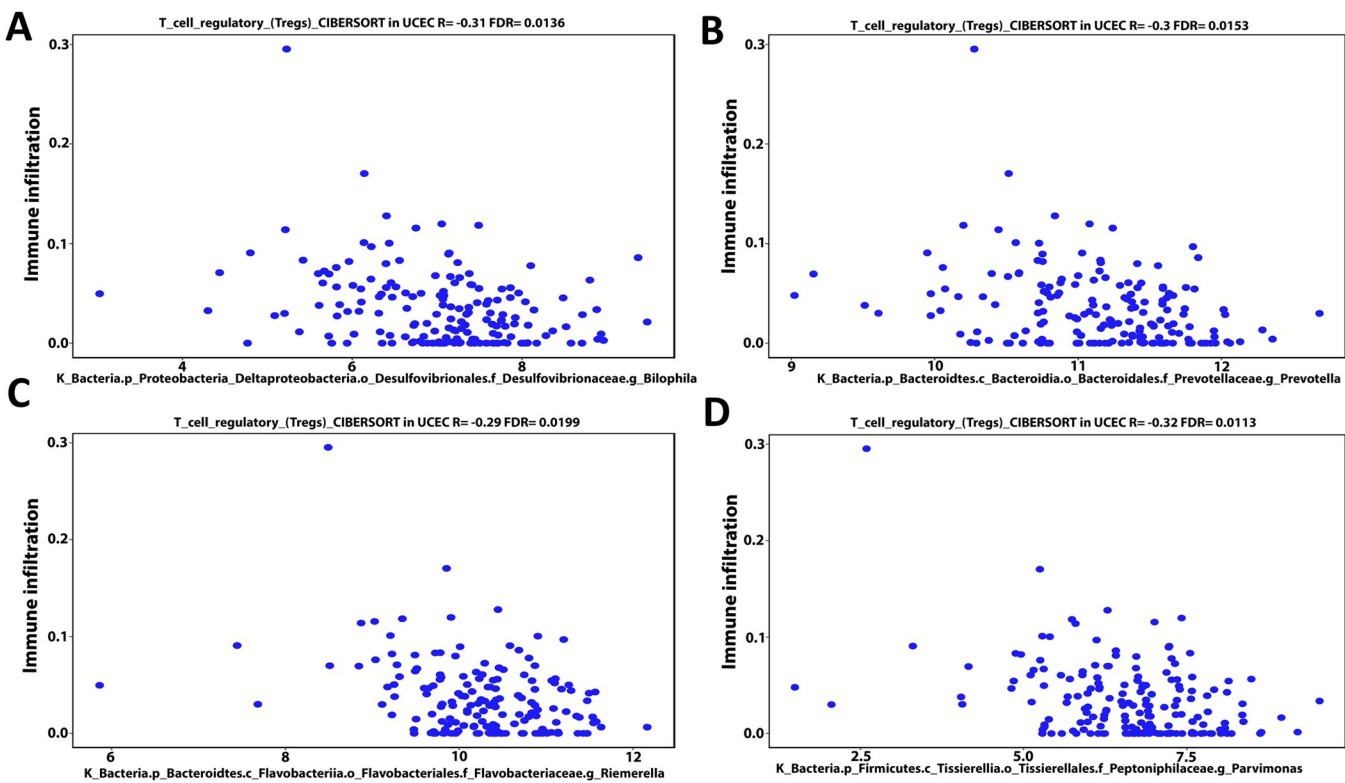

**Fig 7.** Bioinformatics analysis of the CIBERSORT TCGA database deconvolution revealed that microbiome such as Proteobacteria (Bilophila) (A), Bacteroidetes (Prevotella) (B), Bacteroidetes (Riemerella) (C), and Firmicutes (Parvimonas) (D) have a negative pearson correlation coefficient (-0.31, -0.3, -0.29, and -0.32 respectively) with Tregs invasion in TME of UCEC patients.

abnormal immunological responses to gut bacteria. In this regard, they found that genetic variations linked to IBD and their consequences on the gut microbiota, immunological response, and disease development [23]. A recent pioneering study examined techniques as tumor microbiome prediction models for the prognosis and treatment response of four tumor types: UCEC, breast cancer, cervical squamous cell carcinoma, and sarcoma [24]. Another study looked at pan-cancer and discovered that UCCE, ESCA, STAD, CHOL, OV, THYM, TGCT, and MESO had more significant species of microbial communities within these TME than other cancers [25]. This way is not surprising given that we identified that these mutant genes are common in UCEC patients by analyzing the TCGA database.

The study of thirteen mutant genes demonstrated that they all have a significant function in the orchestrate of immunological invasion within TME. Further, our immune cells intersection result involving thirteen mutant genes and microbiome in UCEC are B cell memory, T cells regulatory (Tregs), and M1 macrophage. In patients with UCEC, the memory B cell signature was associated with improved overall survival and higher hazard ratios [26]. SLC7A11 upregulation is related to a better clinical outcome in UCEC patients and has a negative relationship with T cells regulatory (Tregs) infiltration. It is widely established that lower Tregs expression was associated with greater clinical grade and more significant pathological morphology, and patients in later stages had lower Tregs expression in general. However, the prognosis of UCEC patients [27]. Among thirteen mutant genes, our study revealed also that the top three mutant genes drive the invasion of immune cells into the TME of UCEC patients, including PGLYRP2, OLFM4, and TLR5. In UCEC patients, the percentages of PGLYRP2,

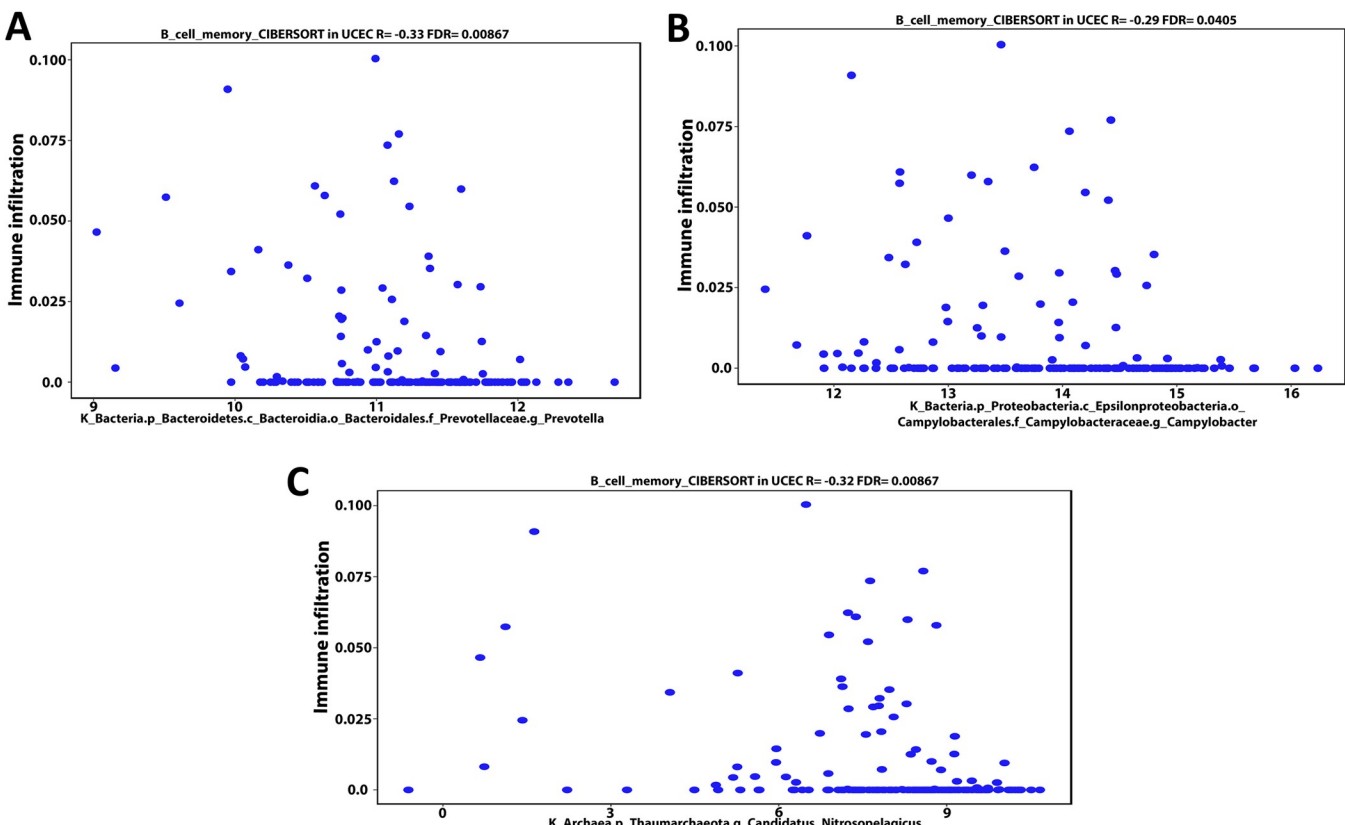

**Fig 8.** Bioinformatics analysis of the deconvolution of the CIBERSORT TCGA database revealed that microbiome such as Bacteroidetes (Prevotella) (A), Proteobacteria (Campylobacter) (B), and Thaumarchaeota (nitrosopelagcus) (C), have a negative pearson correlation coefficient (-0.33, -0.29, and -0.32 respectively) with B cell memory invasion into UCEC patients' TME.

OLFM4, and TLR5 mutations were 4%, 4.7%, and 7.7%, respectively. In the current data, the status of PGLYRP2 was the top of thirteen mutant genes that regulated the invasion of ten immunological cells, including Tregs and B cell memory. PGLYRP2 deletion causes alterations in mice with altered microbiota, which increases the incidence of autism [28]. PGLYRP2 polymorphisms in patients with ulcerative colitis and Crohn's disease are related with gender and/ or age of onset [29]. Tumor-derived PGLYRP2 is well recognized to operate as a possible biomarker for appropriate immune response to many cancers [30]. In wild-type mice, PGLYRP2 decreases Th17 cell overactivation by enhancing regulatory T cell recruitment, thus protecting the skin from excessive inflammation [31]. This is not surprising given that mutant PGLYRP2 in UCEC patients exhibited a lower amount of Tregs than wild type. The status of OLFM4 was the second of thirteen mutant genes that governed the invasion of eight immune cells from UCEC patients, including T cells regulatory (Tregs) and M1 macrophages. A recent study reveals that the presence of OLFM4 in the jejunum is associated with intestinal dysbiosis [32]. Systemic OLFM4 deletion increases colon and prostate carcinogenesis [33, 34]. Here, TLR5 status was shown to coordinate the invasion of seven immune cells, including M1 macrophages. The TLR5 SNP rs5744168 was linked to the incidence of sporadic breast cancer [35]. Microbiota-TLR5 interactions in mice modulate systemic inflammation and cancer, which may be important for patients with breast and OV [36]. The phenotypic M1-like is often defined as proinflammatory and is generated by a Toll like receptor (TLR) [37].

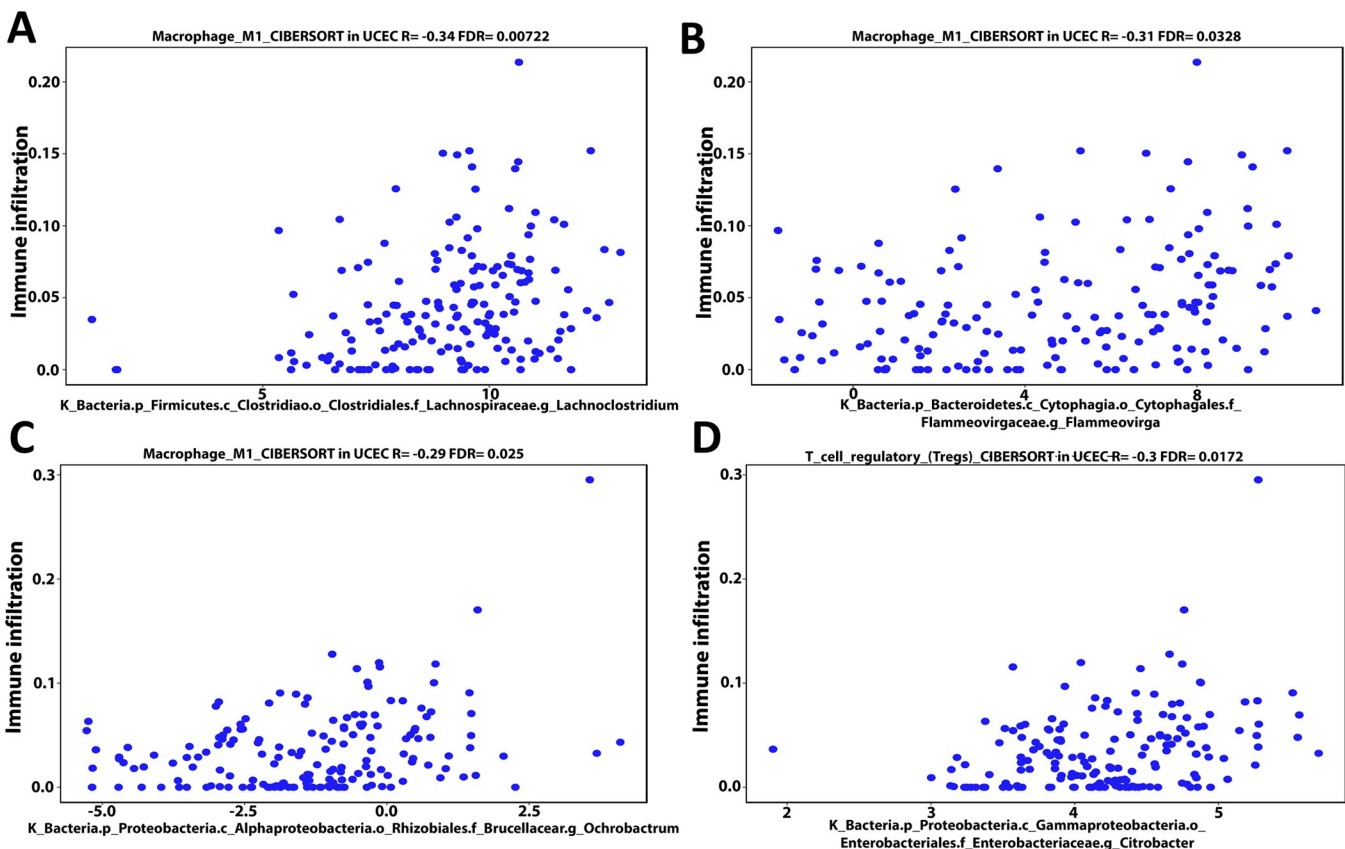

**Fig 9.** Firmicutes (Lachnoclostridium) (A) and Bacteroidetes (Flammeovirga) (B) demonstrate a positive pearson correlation coefficient (0.34 and 0.31) with M1 macrophage invasion, according to bioinformatics analysis of the deconvolution of the CIBERSORT TCGA database. Furthermore, Proteobacteria (Ochrobactrum) (C) and Proteobacteria (Citrobacter) (D) had a positive pearson correlation coefficient (0.29 and 0.3) with Treg migration into UCEC TME.

Our results highlighted that Proteobacteria (bilophila), Bacteroides (Prevotella) (Riemerella), and Firmicutes (parvimonas) had negative relationships with Tregs abundance in UCEC patients. These findings support a prior study that looked at the pathophysiological significance of proteobacteria in asthmatic patients through immunological responses [38]. The change of bacteroides in the gut microbiome is associated to obesity via enhancing immune cell invasion [39]. Bacteroidales had the most consistent microbiome changes in all Pglyrp-deficient animals. Prevotella falsenii was shown to be more prevalent in dextran sulfate sodium-induced colitis [40]. Firmicutes were shown to be considerably lower in the gut microbiota of people suffering from depression. In addition, Actinobacteria, Bacteroidetes, Firmicutes, Fusobacteria, and Proteobacteria account for 96–99% of the oral microbiome, and these microorganisms modulate immunity via Tregs and Th17 cells [41]. Proteobacteria (campylobacter), Bacteroides (Prevotella), and Thaumarchaeota (nitrosopelagicus) were also shown to have negative links with B cell memory in UCEC patients. Moreover, Proteobacteria (ochrobactrum) and (citrobacter) have positive associations with Tregs cells. Several studies have found that Prevotella and Bacteroides species have cross-talk with human colorectal cancer via interleukin-9 [42]. Campylobacter jejuni 81–176, a human clinical isolate, causes colorectal cancer via altering microbial components and transcriptome responses that are linked to the production of cytolethal distending toxin [43].

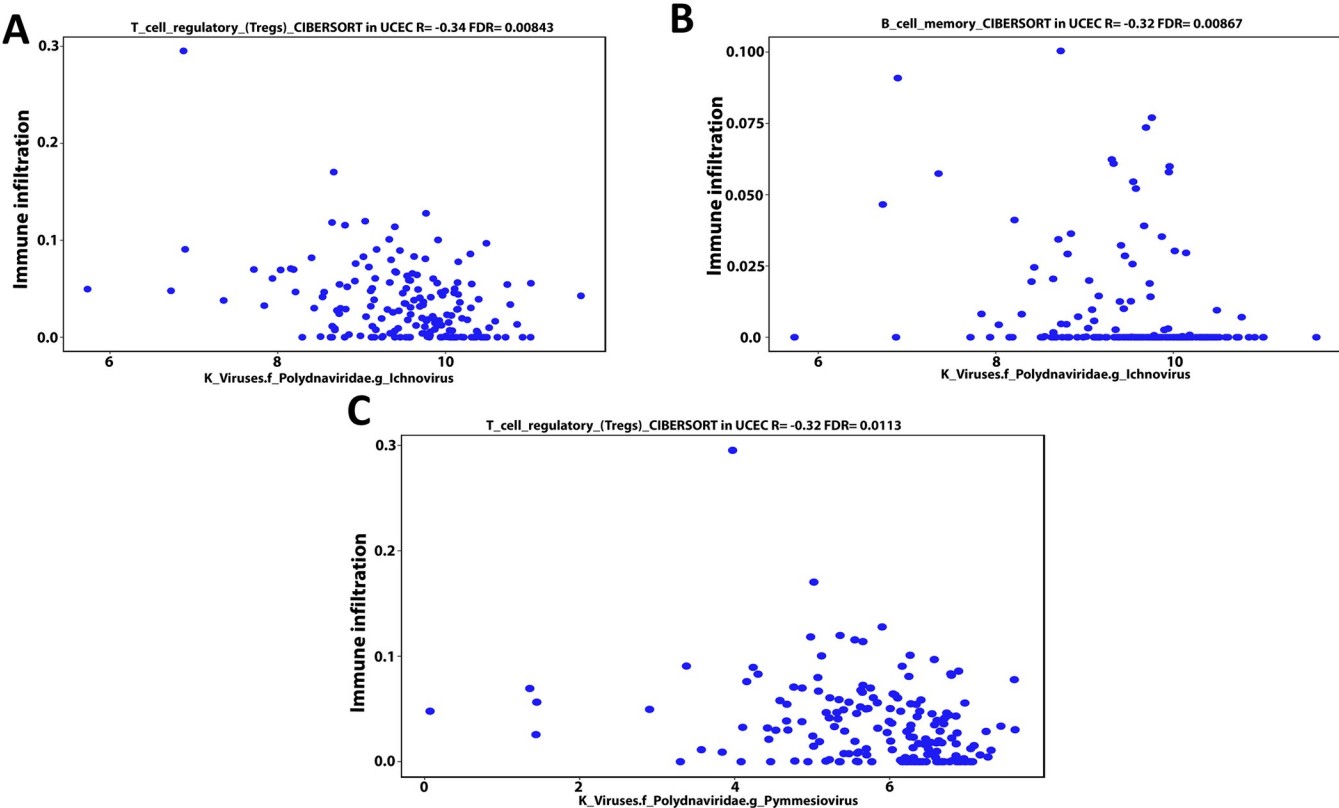

**Fig 10.** According to a bioinformatics study of the CIBERSORT TCGA database deconvolution, microbiome such as Polydnaviridae virus (Ichnovirus) had a negative Pearson correlation coefficient (-0.34 and -0.32) with Tregs (A) and B cell memory (B) invasion into TME. Furthermore, a Pearson association (-0.32) was found between the Phycodnaviridae virus (Prymnesiovirus) and Treg infiltration (C).

On the other hand, over the last several decades, the discovery and characterization of oncoviruses have been at the forefront of biomedical research. It has generated vital insights into fundamental cell biology and cancer pathways [44]. Ichnovirus particles are produced in the female wasp reproductive tract. It contributes significantly to the success of parasitism by parasitoid wasps by offering a variety of instruments for subverting the host immune system and regulating its growth and development [45]. Our findings showed for the first time that viruses such as Polydnaviridae virus (Ichnovirus) have negative correlations with Tregs cell invasion and B cell memory in UCEC. Similarly, the Phycodnaviridae virus (Prymnesiovirus) shows a negative relationship with Tregs cells.

Emerging research suggests that the complex interplay between gut microbiota and the immune system has a significant impact on human and mouse health and illness. In this context, bacteria play a crucial role in training and developing the host immune system, which controls gut microbiota diversity and function in order to maintain homeostasis. Studies have shown that the gut microbiota influences B cell development in both healthy and diseased conditions [45]. In addition, Tregs dysfunction autoimmune disorders include monogenic primary immune deficiencies such as immune dysregulation polyendocrinopathy, enteropathy, X-linked inheritance syndrome, and polygenic autoimmune diseases such as multiple sclerosis, inflammatory bowel disease, and food allergies. These disorders are related with aberrant microbiomes in the small intestine and colon. Some conditions obviously improve with microbial alteration therapy, including probiotics and fecal microbiota transplantation [45]. This is

not unexpected given that our investigation revealed that various bacteria and viruses have been connected to the invasion of immune cells, including B cell memory and T cell regulatory (Tregs), into the TME of UCEC patients.

## 5- Conclusion

We noticed in this study that the common mutant genes that induce abnormal gut flora are also seen and plentiful in UCEC patients. In this regard, we emphasized the potential relationships between thirteen mutant genes causing aberrant gut flora and the invasion of various immune cells inside these patients' TME. Our research found that the top three mutant genes, including PGLYRP2, OLFM4, and TLR5, cause immune cells infiltration into the TME of UCEC patients. Further, our study indicates that certain bacteria and viruses are implicated in the infiltration of immune cells into UCEC patients' TME. Therefore, these findings give light on an essential area for future research into developing innovative immunizations against bacteria or viruses as immunotherapy for UCEC patients.

## Supporting information

**S1 Fig. The TCGA database was used to investigate how the status of PGLYRP1, AICDA, and IKZF1 influences immune cell invasion with TME in UCEC patients.** The PGLYRP1 status affect T cell gamma delta infiltration in UCEC patients (A). Whereas AICDA status affect M1 macrophage infiltration (B) and myeloid dendritic cell activation (C). M1 macrophages (D), M2 macrophages, myeloid dendritic cell active (E), myeloid dendritic cell resting (F), T cell follicular helper (H), and T cell gamma delta (I) infiltration are all affected by IKZF1 status.
(TIF)

**S2 Fig. The TCGA database was used to investigate how the presence of PGLYRP1 influences immune cell invasion with TME in UCEC patients.** PGLYRP1 status orchestrates T cell CD8$^+$ (A), CD4$^+$ memory activated (B), CD4$^+$ memory resting (C), Tregs (D), B cell memory (E), M1 macrophages (F), Myeloid dendritic cell activated (G), mast cell activated (H), T cell follicular helper (I), and T cell gamma delta (J) invasion in UCEC patients.
(TIF)

**S3 Fig. The TCGA database was utilized to evaluate how the status of PGLYRP3 and PGLYRP4 impact immune cell invasion with TME in UCEC patients.** The PGLYRP3 status influences cell T-cell follicular helper infiltration in UCEC patients (A). In UCEC patients, PGLYRP4 status affects T cell CD8$^+$ (B), CD4$^+$ memory activated (C), Tregs (D), M1 macrophages (F), myeloid dendritic cell activated (G), T cell follicular helper (H), and T cell gamma delta (I) invasion.
(TIF)

**S4 Fig. The TCGA database was used to investigate how REG3G status influences immune cell invasion with TME in patients with UCEC.** T cell CD8$^+$ (A), CD4$^+$ memory activated (B), CD4$^+$ memory resting (C), M1 macrophages (D), Myeloid dendritic cell active (E), and myeloid dendritic cell resting (F) invasion is controlled by REG3G status in UCEC patients.
(TIF)

**S5 Fig. The TCGA database was used to assess how MYD88 and PYCARD status affect immune cell invasion with TME in UCEC patients.** Tregs (A), monocytes (B), M1 macrophages (C), myeloid dendritic cell activated (D), T cell follicular helper (E), and T cell gamma delta (F) infiltration in UCEC patients are influenced by MYD88 status. PYCARD status

influences CD4$^+$ memory activated (G), M1 macrophage (H), and T cell gamma delta (I) invasion in UCEC patients.
(TIF)

**S1 File. The intersection result between immune cells regulated by the thirteen mutant genes found in our current proposal and which immune cells in UCEC patients exhibited a positive or negative Pearson correlation coefficient with the microbiome.**
(CSV)

## Acknowledgments

The author would like to acknowledge the Deanship of Research and Entrepreneurship at Shaqra University for supporting this work

## Author Contributions

**Conceptualization:** Samia S. Alkhalil, Amr Ahmed El-Arabey.

**Data curation:** Samia S. Alkhalil, Taghreed N. Almanaa, Mohnad Abdalla, Amr Ahmed El-Arabey.

**Formal analysis:** Samia S. Alkhalil, Amr Ahmed El-Arabey.

**Funding acquisition:** Samia S. Alkhalil, Amr Ahmed El-Arabey.

**Investigation:** Samia S. Alkhalil, Amr Ahmed El-Arabey.

**Methodology:** Samia S. Alkhalil, Amr Ahmed El-Arabey.

**Project administration:** Samia S. Alkhalil, Amr Ahmed El-Arabey.

**Resources:** Samia S. Alkhalil, Amr Ahmed El-Arabey.

**Software:** Samia S. Alkhalil, Mohnad Abdalla, Amr Ahmed El-Arabey.

**Supervision:** Samia S. Alkhalil, Amr Ahmed El-Arabey.

**Validation:** Samia S. Alkhalil, Taghreed N. Almanaa, Raghad A. Altamimi, Mohnad Abdalla, Amr Ahmed El-Arabey.

**Visualization:** Samia S. Alkhalil, Taghreed N. Almanaa, Raghad A. Altamimi, Mohnad Abdalla, Amr Ahmed El-Arabey.

**Writing – original draft:** Samia S. Alkhalil, Amr Ahmed El-Arabey.

**Writing – review & editing:** Samia S. Alkhalil, Taghreed N. Almanaa, Raghad A. Altamimi, Amr Ahmed El-Arabey.

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
