## [Decision Letter · Decision Letter 0]

26 Jun 2024

PONE-D-24-16304Microbiomes and uterine corpus endometrial cancer: A novel crosstalk between abnormal gut microbiota and immunotherapyPLOS ONE

Dear Dr. El-Arabey,

Thank you for submitting your manuscript to PLOS ONE. After careful consideration, we feel that it has merit but does not fully meet PLOS ONE’s publication criteria as it currently stands. Therefore, we invite you to submit a revised version of the manuscript that addresses the points raised during the review process.

We look forward to receiving your revised manuscript.

Kind regards,

Elham Samami

Academic Editor

PLOS ONE

Journal Requirements:

3. Thank you for stating the following in your Competing Interests section: ""The authors declare that they have no known competing financial interests or personal relationships that could have appeared to influence the work reported in this paper.""

Additional Editor Comments:

Dear Authors,

We are pleased to inform you that your manuscript has been accepted with minor revisions. We would like to bring the following points to your attention for improvement:

**Title Adjustment**: It is necessary to modify the title to better reflect the computational aspects of your work. This will help highlight the methodology and analytical approaches used in your study.**Elaboration on Databases**: Your study heavily relies on existing databases such as TCGA, TIMER2.0, and Cancer-mbQTL. Please provide a more detailed description in the Methods section, specifically elaborating on how these databases were linked and utilized to support your research findings.**Statistical Analysis**: To strengthen your conclusions, it is recommended to incorporate inferential statistical tests. These tests should assess the significance of differences in mutation frequencies and immune cell infiltration, providing a more robust foundation for your findings.**Correlation Analysis**: The use of Pearson correlation assumes a linear relationship between variables, which may not accurately capture the complex, nonlinear interactions inherent in biological systems. Consider discussing alternative methods that might better represent these interactions.**Model Validation**: To prevent overfitting, we recommend implementing cross-validation techniques, such as k-fold cross-validation, and using regularization methods like LASSO or ridge regression. Additionally, independent validation with separate datasets is advised to ensure the robustness of your findings and to avoid large dataset overfitting.**Broader Context in Discussion**: The Discussion section consequently should include a more comprehensive analysis of  microbial interactions and their potential effects on the tumor microenvironment (TME). This will provide a deeper context for your results and implications.

We look forward to receiving your revised manuscript. Should you have any questions or need further clarification, please do not hesitate to contact us.

Reviewers' comments:

Reviewer's Responses to Questions

**Comments to the Author**

1. Is the manuscript technically sound, and do the data support the conclusions?

Reviewer #1: Yes

Reviewer #2: Partly

2. Has the statistical analysis been performed appropriately and rigorously? 

Reviewer #1: No

Reviewer #2: N/A

3. Have the authors made all data underlying the findings in their manuscript fully available?

Reviewer #1: Yes

Reviewer #2: Yes

4. Is the manuscript presented in an intelligible fashion and written in standard English?

Reviewer #1: Yes

Reviewer #2: Yes

5. Review Comments to the Author

Reviewer #1: It's needed for title to point towrad the computational nature of this work.

The study heavily relies on existing databases such as TCGA, TIMER2.0, and Cancer-mbQTL and it is needed to elaborate more in methods ,what was the process of linking the function of these databases.

Incorporating inferential statistical tests to assess the significance of differences in mutation frequencies and immune cell infiltration would provide more robust conclusions.

Pearson correlation assumes a linear relationship between variables, which may not accurately represent the complex, nonlinear interactions in biological systems.

Implementing cross-validation techniques, such as k-fold cross-validation, and using regularization methods (e.g., LASSO, ridge regression) can help prevent overfitting. Independent validation with separate datasets is also recommended to avoid large dataset overfitting.

Broader microbial interactions and their effects on the TME is needed to be discussed in discussion.

Reviewer #2: Please use the space provided to explain your answers to the questions above. You may also include additional comments for the author, including concerns about dual publication, research ethics, or publication ethics. (Please upload your review as an attachment if it exceeds 20,000 characters) (Limit 200 to 20000 Characters)

6. PLOS authors have the option to publish the peer review history of their article (what does this mean?). If published, this will include your full peer review and any attached files.

Reviewer #1: No

Reviewer #2: No

---

## [Author Response · Author response to Decision Letter 0]

2 Sep 2024

The academic Editor, 9 th August, 2024

Plos One

Subject: (Revised manuscript (Ref. PONE-D-24-16304) and our response to Reviewer’s comments). 

Dear Prof. Elham Samami

 With great enthusiasm, we are submitting our revised manuscript entitled, “Interactions between microbiota and uterine corpus endometrial cancer: A bioinformatic investigation of potential immunotherapy” by Samia S. Alkhalil 1* , Taghreed N Almanaa 2, Raghad A Albedair 3, Mohnad Abdalla4 and Amr Ahmed El-Arabey5*, for consideration publication in Plos One. We are very grateful for the constructive and positive feedback of the reviewers and editor. Following the reviewers’ suggestions, we have revised our work and manuscript against the reviewer’s comment. Please see the below document titled “Reviewers’ comments/questions and their answers”, in which we have included detailed point-to-point responses. We believe that we have fully addressed all concerns raised in the original submission, and the current revised manuscript has been substantially strengthened and improved for possible publication in Plos One. We sincerely hope this revised manuscript would satisfy both the reviewers and the editor and would be happy to answer any additional questions anytime. 

Thank you for your valuable time in reviewing our work, and look forward to your reply.

Looking forward to hearing from you.

Kind Regards,

Amr Ahmed El-Arabey (Corresponding author)

Email: ph.amrcapa@gmail.com

Reviewers’ comments/questions and their answers

1. Title Adjustment: It is necessary to modify the title to better reflect the computational aspects of your work. This will help highlight the methodology and analytical approaches used in your study.

Response: Thank you so much for your positive feedbacks. As suggested by the reviewer, we revised our manuscript (Red discoloration in the manuscript):

Interactions between microbiota and uterine corpus endometrial cancer: A bioinformatic investigation of potential immunotherapy”

2. Elaboration on Databases: Your study heavily relies on existing databases such as TCGA, TIMER2.0, and Cancer-mbQTL. Please provide a more detailed description in the Methods section, specifically elaborating on how these databases were linked and utilized to support your research findings.

Response: Done: (Red discoloration in the manuscript):

3. Statistical Analysis: To strengthen your conclusions, it is recommended to incorporate inferential statistical tests. These tests should assess the significance of differences in mutation frequencies and immune cell infiltration, providing a more robust foundation for your findings.

Response: In the current proposal we used the Cancer-mbQTL database to discover immune cells from CIBERSORT deconvolution of TCGA data that correlate with the abundance levels of microbes in tumor tissues of UCEC patients (designated as cancer microbiome quantitative trait loci, cancer-mbQTL). This comprehensive mbQTL database aids in successfully evaluating the influence of gene mutations on cancer microbiome characteristics, establishing a new paradigm for understanding the function of risk genetic mutations in human malignancies.

4. Correlation Analysis: The use of Pearson correlation assumes a linear relationship between variables, which may not accurately capture the complex, nonlinear interactions inherent in biological systems. Consider discussing alternative methods that might better represent these interactions.

Response: The Pearson correlation assumes that the relationship between variables is linear. If the relationship is not linear, Pearson’s correlation may not accurately represent the association. In this regard, we used Spearman correlation to determine the relationship between microbial abundance and immune cell infiltration in uterine corpus endometrial cancer using CIBERSORT method. Associations with FDR < 0.05 were deemed statistically significant. Kraken-derived normalized microbial abundances of TCGA patients were collected from the Cancer Microbiome. This comprehensive mbQTL database aids in successfully evaluating the influence of gene mutations on cancer microbiome characteristics, establishing a new paradigm for understanding the function of risk genetic mutations in human malignancies.

5. Model Validation: To prevent overfitting, we recommend implementing cross-validation techniques, such as k-fold cross-validation, and using regularization methods like LASSO or ridge regression. Additionally, independent validation with separate datasets is advised to ensure the robustness of your findings and to avoid large dataset overfitting.

Response: In the current proposal, we provide CIBERSORT, a technique for defining the cell composition of complex tissues using gene expression data. When applied to enumerating hematopoietic subsets in RNA mixes from fresh, frozen, and fixed tissues, including solid tumors, CIBERSORT outperformed other approaches in terms of noise, unknown mixture content, and closely related cell types. CIBERSORT should facilitate large-scale investigation of RNA mixtures for biological biomarkers and medicinal targets. Besides, there are limited bioinformatic database of microbiota to introduce validation with separate database. 

Reference: Newman AM, Liu CL, Green MR, Gentles AJ, Feng W, Xu Y, Hoang CD, Diehn M, Alizadeh AA. Robust enumeration of cell subsets from tissue expression profiles. Nat Methods. 2015 May;12(5):453-7. doi: 10.1038/nmeth.3337.

6. Broader Context in Discussion: The Discussion section consequently should include a more comprehensive analysis of microbial interactions and their potential effects on the tumor microenvironment (TME). This will provide a deeper context for your results and implications.

Response: Done: (Red discoloration in the manuscript):

---

## [Editor Report · Decision Letter 1]

24 Sep 2024

PONE-D-24-16304R1Interactions between microbiota and uterine corpus endometrial cancer: A bioinformatic investigation of potential immunotherapyPLOS ONE

Dear Dr. El-Arabey,

Thank you for submitting your manuscript to PLOS ONE. After careful consideration, we feel that it has merit but does not fully meet PLOS ONE’s publication criteria as it currently stands. Therefore, we invite you to submit a revised version of the manuscript that addresses the points raised during the review process.

We look forward to receiving your revised manuscript.

Kind regards,

Elham Samami

Academic Editor

PLOS ONE

Journal Requirements:

Additional Editor Comments:

Please provide a new draft of the paper with an open review tracker that highlights the recent changes made since the previous submission.

---

## [Author Response · Author response to Decision Letter 1]

25 Sep 2024

The academic Editor 

Plos One

Subject: (Revised manuscript (Ref. PONE-D-24-16304) and our response to Reviewer’s comments). 

Dear Prof. Elham Samami

 With great enthusiasm, we are submitting our revised manuscript entitled, “Interactions between microbiota and uterine corpus endometrial cancer: A bioinformatic investigation of potential immunotherapy” by Samia S. Alkhalil 1* , Taghreed N Almanaa 2, Raghad A Albedair 3, Mohnad Abdalla4 and Amr Ahmed El-Arabey5*, for consideration publication in Plos One. We are very grateful for the constructive and positive feedback of the reviewers and editor. Following the reviewers’ suggestions, we have revised our work and manuscript against the reviewer’s comment. Please see the below document titled “Reviewers’ comments/questions and their answers”, in which we have included detailed point-to-point responses. We believe that we have fully addressed all concerns raised in the original submission, and the current revised manuscript has been substantially strengthened and improved for possible publication in Plos One. We sincerely hope this revised manuscript would satisfy both the reviewers and the editor and would be happy to answer any additional questions anytime. 

Thank you for your valuable time in reviewing our work, and look forward to your reply.

Looking forward to hearing from you.

Kind Regards,

Amr Ahmed El-Arabey (Corresponding author)

Email: ph.amrcapa@gmail.com

Reviewers’ comments/questions and their answers

1. Title Adjustment: It is necessary to modify the title to better reflect the computational aspects of your work. This will help highlight the methodology and analytical approaches used in your study.

Response: Thank you so much for your positive feedbacks. As suggested by the reviewer, we revised our manuscript (Red discoloration in the manuscript):

Interactions between microbiota and uterine corpus endometrial cancer: A bioinformatic investigation of potential immunotherapy”

2. Elaboration on Databases: Your study heavily relies on existing databases such as TCGA, TIMER2.0, and Cancer-mbQTL. Please provide a more detailed description in the Methods section, specifically elaborating on how these databases were linked and utilized to support your research findings.

Response: Done: (Red discoloration in the manuscript):

3. Statistical Analysis: To strengthen your conclusions, it is recommended to incorporate inferential statistical tests. These tests should assess the significance of differences in mutation frequencies and immune cell infiltration, providing a more robust foundation for your findings.

Response: In the current proposal we used the Cancer-mbQTL database to discover immune cells from CIBERSORT deconvolution of TCGA data that correlate with the abundance levels of microbes in tumor tissues of UCEC patients (designated as cancer microbiome quantitative trait loci, cancer-mbQTL). This comprehensive mbQTL database aids in successfully evaluating the influence of gene mutations on cancer microbiome characteristics, establishing a new paradigm for understanding the function of risk genetic mutations in human malignancies.

4. Correlation Analysis: The use of Pearson correlation assumes a linear relationship between variables, which may not accurately capture the complex, nonlinear interactions inherent in biological systems. Consider discussing alternative methods that might better represent these interactions.

Response: The Pearson correlation assumes that the relationship between variables is linear. If the relationship is not linear, Pearson’s correlation may not accurately represent the association. In this regard, we used Spearman correlation to determine the relationship between microbial abundance and immune cell infiltration in uterine corpus endometrial cancer using CIBERSORT method. Associations with FDR < 0.05 were deemed statistically significant. Kraken-derived normalized microbial abundances of TCGA patients were collected from the Cancer Microbiome. This comprehensive mbQTL database aids in successfully evaluating the influence of gene mutations on cancer microbiome characteristics, establishing a new paradigm for understanding the function of risk genetic mutations in human malignancies.

5. Model Validation: To prevent overfitting, we recommend implementing cross-validation techniques, such as k-fold cross-validation, and using regularization methods like LASSO or ridge regression. Additionally, independent validation with separate datasets is advised to ensure the robustness of your findings and to avoid large dataset overfitting.

Response: In the current proposal, we provide CIBERSORT, a technique for defining the cell composition of complex tissues using gene expression data. When applied to enumerating hematopoietic subsets in RNA mixes from fresh, frozen, and fixed tissues, including solid tumors, CIBERSORT outperformed other approaches in terms of noise, unknown mixture content, and closely related cell types. CIBERSORT should facilitate large-scale investigation of RNA mixtures for biological biomarkers and medicinal targets. Besides, there are limited bioinformatic database of microbiota to introduce validation with separate database. 

Reference: Newman AM, Liu CL, Green MR, Gentles AJ, Feng W, Xu Y, Hoang CD, Diehn M, Alizadeh AA. Robust enumeration of cell subsets from tissue expression profiles. Nat Methods. 2015 May;12(5):453-7. doi: 10.1038/nmeth.3337.

6. Broader Context in Discussion: The Discussion section consequently should include a more comprehensive analysis of microbial interactions and their potential effects on the tumor microenvironment (TME). This will provide a deeper context for your results and implications.

Response: Done: (Red discoloration in the manuscript):

---

## [Editor Report · Decision Letter 2]

10 Oct 2024

Interactions between microbiota and uterine corpus endometrial cancer: A bioinformatic investigation of potential immunotherapy

PONE-D-24-16304R2

Dear Dr. El-Arabey,

We’re pleased to inform you that your manuscript has been judged scientifically suitable for publication and will be formally accepted for publication once it meets all outstanding technical requirements.

Kind regards,

Elham Samami

Academic Editor

PLOS ONE

---

## [Editor Report · Acceptance letter]

18 Oct 2024

PONE-D-24-16304R2 

PLOS ONE

Dear Dr. El-Arabey, 

I'm pleased to inform you that your manuscript has been deemed suitable for publication in PLOS ONE. Congratulations! Your manuscript is now being handed over to our production team.

Kind regards, 

on behalf of

. Elham Samami 

Academic Editor

PLOS ONE